# Determination of the refractive index of insoluble organic extracts from atmospheric aerosol over the visible wavelength range using optical tweezers

Rosalie H. Shepherd[1, 2], Martin D. King[2], Amelia Marks[2], Neil Brough[3] and Andrew D. Ward[1]

[1]Central Laser Facility, Research Complex, STFC Rutherford Appleton Laboratory, Oxford, OX11 0FA, UK
[2]Department of Earth Sciences, Royal Holloway University of London, Egham, Surrey, TW20 0EX, UK
[3]British Antarctic Survey, High Cross, Madingley Road, Cambridge, CB3 0ET, UK

*Correspondence to*: Martin King (m.king@rhul.ac.uk)

Optical trapping combined with Mie spectroscopy is a new technique used to record the refractive index of insoluble organic material extracted from atmospheric aerosol samples over a wide wavelength range. The refractive index of the insoluble organic extracts was shown to follow a Cauchy equation between 460 to 700 nm for organic aerosol extracts collected from urban (London) and remote (Antarctica) locations. Cauchy coefficients for the remote sample were for the Austral summer and gave the Cauchy coefficients of A = 1.467 and B = 1000 nm$^2$ with a real refractive index of 1.489 at a wavelength of 589 nm. Cauchy coefficients for the urban samples varied with season, with extracts collected during summer having Cauchy coefficients of A = 1.465±0.005 and B = 4625±1200 nm$^2$ with a representative real refractive index of 1.478 at a wavelength of 589 nm, whilst samples extracted during autumn had larger Cauchy coefficients of A=1.505 and B = 600 nm$^2$ with a representative real refractive index of 1.522 at a wavelength of 589 nm. The refractive index of absorbing aerosol was also recorded. The absorption Ångström exponent was determined for wood smoke and humic acid aerosol extract. Typical values of the Cauchy coefficient for the wood smoke aerosol extract were A = 1.541±0.03 and B = 14800±2900 nm$^2$ resulting in a real refractive index of 1.584±0.007 at a wavelength of 589 nm and an absorption Ångström exponent of 8.0. The measured values of refractive index compare well with previous monochromatic or very small wavelength range measurements of refractive index. In general, the real component of the refractive index increases from remote to urban to wood smoke. A one-dimensional radiative-transfer calculation of the top of the atmosphere albedo was applied to model an atmosphere containing a 3 km thick layer of aerosol comprising of pure water, pure insoluble organic aerosol or an aerosol consisting of an aqueous core-with an insoluble organic shell. The calculation demonstrated that the top of the atmosphere albedo increases by 0.01 to 0.04 for pure organic particles relative to water particles of the same size and the top of the atmosphere albedo increases by 0.03 for aqueous core-shell particles as volume fraction of the shell material increases to 25 %.

## 1. Introduction

Atmospheric aerosols affect the radiative balance of our planet (see e.g. Pöschl et al., 2005; Ramanathan et al., 2001; Stocker et al., 2013; Wild et al., 2009). Aerosols directly affect the radiative balance by absorbing or scattering incoming solar radiation (see e.g. Moise et al., 2015), and indirectly through their role as cloud condensation nuclei (see e.g. Breon et al., 2002; Lohmann and Feichter, 2005; Rosenfeld et al., 2008; Charlson et al., 2001). The current understanding of the atmospheric aerosol radiative forcing and the cloud albedo effect is currently regarded as low compared to other radiative effects such as greenhouse gases (see e.g. Stocker et al., 2013; Fuzzi et al., 2005).

Atmospheric aerosols contain a complex mixture of many different chemical compounds with a wide variety of physio-chemical properties (see e.g. Beddows et al., 2004; Cappa et al., 2011; Cai et al., 2016; Cochran et al., 2016). The contrasting properties of the different compounds can result in atmospheric aerosols being coated in a film of organic material (see e.g. Gill et al., 1983; Donaldson and Vaida, 2006). The presence of a film may alter the physical, chemical and optical properties of the cloud droplet or aerosol particle by (a) reducing the rate of evaporation from the droplets or particles (see e.g. Davies et al., 2013; Eliason et al., 2003; Gill et al., 1983; Kaiser et al., 1996; McFiggans et al., 2005), (b) altering the transport of chemicals between gas and liquid phase (see e.g. Donaldson and Anderson, 1999; Donaldson and Valsaraj, 2010), (c) affecting gaseous uptake (see e.g. Enami et al., 2010), (d) reducing the scavenging by larger cloud droplets (see e.g. Andreaea and Rosenfeld, 2008; Feingold and Chuang, 2002) and (e) altering the optical properties of the droplet (see e.g. Donaldson and Vaida, 2006; Li et al., 2011). To quantify the consequences of film formation on atmospheric aerosol particles and cloud droplets, further understanding of aerosol films is urgently required. However, obstacles such as accurately quantifying anthropogenic and natural aerosol emissions (see e.g. Kanakidou et al., 2005) or understanding the hugely varying chemistry of atmospheric aerosols (see e.g. Jacobson and Hansson, 2000) means the current understanding of atmospheric aerosol remains low (see e.g. Stocker et al., 2013; Flores et al., 2014).

Furthering the current understanding of film formation on cloud droplets or aerosols has largely been addressed by replicating the film with atmospheric proxy compounds such as oleic acid (see e.g. King et al., 2004,2009; Jones et al., 2015), methyl oleate (see e.g. Pfrang et al., 2014; Sebastiani et al., 2015), pinonic acid (see e.g. Enami and Sakamoto, 2016), 1,2-dipalmitoyl-*sn*-glycero-3-phosphocholine (DPPC) (see e.g. Thompson et al., 2010), or nonanoic acid (see e.g. Tinel, 2016). However organic material extracted from the environment has only featured in a limited number of studies that investigated the consequences of exposing organic films extracted from atmospheric aerosol and sea-water to ozone (see e.g. Jones et al., 2017), or studied the heterogeneous chemistry of material collected from the surface of the sea (see e.g. Zhou et al. 2014). Research that focus on the sea-surface layer is highly relevant to atmospheric studies owing to sea-surface material often becoming aerosols (see e.g. Blanchard et al., 1964).

The work presented describes the application of the optical trapping technique to measure the refractive index of organic material that may form an organic film on atmospheric aerosol. The organic material was extracted from atmospheric aerosol (using the techniques of Folch and Lee, 1957 and Bligh and Dyer, 1959) and the refractive index of the

aerosol measured through the application of white light scattering in conjunction with Mie spectroscopy (see e.g. Bohren and Huffman, 1983). The values of the refractive index were subsequently used to estimate the change in the top of the atmosphere albedo from radiative transfer calculations that modelled an aqueous aerosol with a thin film of the atmospheric aerosol film material.

5      Determining the refractive index of atmospheric aerosol is paramount to understanding the light scattering and absorption properties of atmospheric aerosol, and hence its contribution to global temperatures. Considerable work has focused on determining the real and imaginary component of the refractive indices of atmospheric aerosols (see e.g. Liu et al., 2013; Wex et al., 2009; Barkey et al., 2007; Meskhidze et al., 2013; Nakayama et al., 2010; and Lang-Yona et al., 2010). The use of morphological dependent resonances in Raman Spectra to determine refractive index at a fixed wavelength has been reported by Lin et al. (1990) and references therein and Miles et al. (2012). The absorbing properties and optical extinction properties of atmospheric aerosols have been extensively studied (see e.g. Zhao et al., 2013). Washenfelder et al., (2013) studied aerosol extinction in the ultra-violet region and Liu et al., (2016) investigated the absorbing properties of brown carbon. In the study presented, wavelength dependent refractive indices were determined for common aerosol types such as organic atmospheric aerosol sourced from urban and remote locations and from wood smoke aerosol. In addition, the wavelength-dependent refractive index of the proxy atmospheric aerosol, humic acid, was determined.

      Application of the optical trapping technique was successfully employed to determine the refractive index of aerosol over a wide wavelength range. The technique allowed the refractive index to be resolved to within 0.015 over a large wavelength range of 460 to 700 nm (see e.g. David et al., 2016; Jones et al., 2013, 2015). Previous studies have determined the refractive index of atmospheric aerosols either over a narrower or monochromatic wavelength range. Flores (2014) determined the refractive index for secondary organic aerosol over the wavelength range of 360 to 420 nm, whilst Lambe (2013), Guyon (2003) and Nakayama (2015) studied the refractive index of aerosols at individual wavelengths.

      The technique also allowed the imaginary component of the refractive index and absorption Ångström exponent to be considered for the wood smoke aerosol extracts and aqueous humic acid aerosol owing to the samples absorbing at smaller wavelengths. The absorption Ångström exponent describes the wavelength dependence of the absorption of light by aerosols (see e.g. Moosmüller et al., 2011). The absorption Ångström exponent of atmospheric aerosols has been shown to be sensitive to wavelength (see e.g. Chakrabarty, 2010; Lewis, 2008), chemical composition (see e.g. Ajtai et al., 2011; Chakrabarty et al., 2010; Flowers et al., 2010; Park and Yu, 2016; Russell et al., 2009; Sandradewi et al., 2008), morphology (see e.g. Liu et al., 2008; Utry et al., 2014) and size (see e.g. Utry et al., 2014; Gyawali et al., 2009).

## 2. Materials and Methods

To optically trap aerosol extracts, aerosol was collected from the atmosphere by pulling air through a pre-combusted quartz filter using an air pump. The aerosol material was then extracted from the filter and transferred to isopropanol from which airborne aerosols could by generated by ultrasonic nebulisation. Additionally, a commercial sample of humic acid in an

aqueous solution was studied as an aerosol. The airborne aerosols were optically trapped and illuminated with white light, the backscattered white light was collected to generate a Mie spectrum of scattered light intensity as a function of wavelength. From the Mie spectrum, the wavelength dependent refractive index and radius could be determined by replication with Mie calculations.

## 2.1. Collection of aerosol extracts

Three types of atmospheric aerosol were sampled to demonstrate the technique. The samples were chosen to represent (a) a "dirty" sample from an urban environment, (b) a "clean" sample from a remote location and (c) samples with strong light absorbing properties. Wood smoke aerosol extract and aqueous humic acid aerosol were chosen for this purpose.

The urban sample was collected from the campus of Royal Holloway, University of London. The proximity of Central London (30 km), major motorways (M25, M40 and M4) and the large international airport Heathrow (8 km) mean the sample has been categorised as urban for the purposes of the study. The urban atmospheric aerosol extracts were collected over 30-day periods and combined in the extraction process to allow seasonal (spring, summer, autumn and winter) analysis. Combining the urban aerosol extracts into seasons ensured enough material was present to create airborne aerosols and to optically trap, in a relatively inefficient process, as described in Sect. 2.2.

The remote atmospheric aerosol extract was collected at the Halley Clean Air Sector Laboratory operated by the British Antarctic Survey (see e.g. Jones et al., 2008). The sample was collected over the Antarctic summer of 2015 for 60 consecutive days. Antarctica is situated far from human populated areas, and therefore the sample has been called remote aerosol for the purpose of the study.

The wood smoke was collected over a six-hour time period from the smoke plume of a flaming log fire burning in a domestic wood burner. The firewood used in the wood burner was sourced from wild cherry trees. Two wood smoke samples were collected from two separate fires, and were labelled extract A and extract B.

To determine the contamination for each sample, corresponding analytical filter blanks were collected. To ensure all possible contamination sources were accounted for, the remote and wood smoke analytical blanks travelled to the field sites, and back, with the collected sample.

The urban and wood smoke aerosols were sampled by using an air pump with a flow of 30 L min$^{-1}$ through clean stainless steel pipelines into a PFA (perfluoroalkoxy) Savillex filter holder, whilst remote aerosols were sampled from Antarctic ambient air by using a short length of quarter inch O.D. PFA tubing at a flow rate of 20 L min$^{-1}$ onto a filter holder using a Staplex low volume air sampler (Model VM-4). All filter holders contained pre-combusted quartz filters (SKC Ltd.) with a diameter of 47 mm. The air was pulled through the filter for a known time period, after which the sample and filter were frozen in the dark at -18 °C until the sample could be extracted from the filter (typically days).

After sample collection, the filter holder was dissembled in a clean glove box to prevent contamination. The filter encased in the filter holder was cut into two on clean PFA blocks with a stainless-steel blade: one half was for analysis, the other for reference. The filter was placed in a glass conical flask with 10 ml of chloroform (Sigma-Aldrich, 0.5 to 1 %

ethanol as stabiliser) and 10 ml of ultrapure water (> 18 MΩ m$^{-1}$), and the mixture sonicated for 10 minutes. After sonication, the mixture was filtered through pre-combusted quartz filters to remove filter debris. The filter debris with un-extracted material was discarded. Sequentially, the filtrate was poured into a glass separating funnel and the chloroform layer drawn off. The atmospheric aerosol extract studied was soluble in chloroform. The chloroform was removed from the atmospheric

aerosol extract by evaporation under dry nitrogen. Once all the chloroform had been removed, 2 ml of isopropanol (Sigma-Aldrich, purity ≥ 99.8 %) was added to the sample and the sample was stored in the dark at -18 °C until use. All instrumentation used in the collection and extraction of the atmospheric aerosol extract was cleaned with ultrapure water and chloroform before use. The sonication in the extraction process was not found to change the Langmuir-isotherm of atmospheric material at the air-water interface.

Every sample extracted from the atmosphere had a corresponding analytical blank to check for potential contamination of the filters. The analytical blanks for each sample were extracted following the same procedure above.

In addition to atmospheric aerosol extracts, aqueous humic acid aerosol (Sigma-Aldrich, humic acid sodium salts, STBD5313V) was studied. The nebulised aqueous humic acid solution was prepared at a concentration of 0.0005 g cm$^{-3}$.

## 2.2. Optical trapping of the aerosols

A vertically aligned, counter propagating optical trap was used to optically catch and levitate the aerosols. The optical trapping description is described in full by Jones et al., (2013), however a brief description will be given here. The optical trap consisted of two laser beams that were fibre coupled from a 1064 nm continuous wave Nd:YAG laser (Laser Quantum). The laser beams passed through beam expansion optics to two vertically opposed microscope objectives (Mitutoyo M Plan Apo 50 × NA 0.42). An aluminium sample cell (volume 38.4 cm$^3$) was placed between the two microscope objectives and

used to contain the aerosols. Borosilicate coverslip windows allowed the focused laser beams to pass into the sample cell and form the optical trap. The aerosol could be held for more than 24 hours once trapped (see e.g. Rkiouak et al., 2014). The relative humidity and the temperature of the trapping environment were held at ambient conditions (30 % relative humidity and 20 °C).

A nebuliser (ultrasonic nebuliser, Omron) was used to create aerosols from either the atmospheric aerosol extract in

isopropanol or the aqueous humic acid solution, (see e.g. King et al., 2008). Inlet and exhaust ports allowed aerosol delivery into the sample cell. The solvent for the atmospheric aerosol extracts was exchanged from chloroform (removed by blowing down with nitrogen) to isopropanol prior to optically trapping as chloroform was found to be unsuitable for ultrasonic nebulisation. Isopropanol was added in the volume ratio 5:1 isopropanol to atmospheric extract, and this mixture was nebulised to deliver airborne droplets. Isopropanol evaporated from the aerosol both during transit and immediately upon

capture in the optical trap.

## 2.3. Data analysis

The optically trapped aerosol droplet was illuminated with white light and the elastically backscattered light was collected over a 25° cone angle as a function of wavelength by an objective lens. Further optics described in Jones et al., (2013) focussed the light onto a spectrometer (Acton SP2500i). The resulting spectrum was a function of light intensity versus wavelength and will be called a Mie spectrum henceforth. The Mie spectrum covered the wavelength range 460 to 700 nm, with a resolution of 0.06 nm per pixel. The measured Mie spectrum was simulated using the computational methods of Bohren and Huffman (1983); integrating over a cone angle of backscattered light of 25°. The simulated Mie spectrum was calculated as a function of wavelength, with the wavelength dependence of the refractive index being described by a Cauchy equation:

$$n = A + \frac{B}{\lambda^2} + \frac{C}{\lambda^4} , \qquad\qquad (1)$$

allowing determination of both size and a precise estimation of refractive index as a function of wavelength. The variables in Eq. (1) represent the refractive index, $n$, Cauchy empirical constants $A$, $B$ and $C$, and wavelength, $\lambda$. The values of the three empirical constants and the radius of the trapped aerosol were iterated until a good comparison was achieved between the simulated and the experimentally obtained Mie spectrum. Typically, the radius of droplet was fixed and the values of A, B, and C varied until a good fit between measured and simulated Mie spectra was achieved by simple comparison (inspection) of peak, trough and inflection point positions. The value of the radius was then iterated through a series of radii with optimization of the values of A, B, and C as a function of wavelength repeated at each radius (in steps of $\Delta$A= 0.0001, $\Delta$B=25 nm$^2$, and $\Delta$C= 5×10$^6$ nm$^4$). Thus, a qualitative grid search was performed over parameter space. Parameter space was A varying from 1.3 to 1.7, B from 0 to 20,000 nm$^2$ and C from 0 to 1×10$^9$ nm$^4$. When simulating the Mie spectra of atmospheric aerosol extracts, small adjustment of the empirical constant, $C$, did not alter the simulated Mie spectrum noticeably and this value was therefore held at zero. An estimation of the uncertainty in the derived values of the refractive index and radius of the aerosol could be determined by variation of radius, A and B in turn and comparing the experimental and simulated spectra. The value of the radius was between 0.4 to 1.5 μm typically. The imaginary component of the refractive index was considered only after the grid search for the real component of refractive index of the woodsmoke and humic acid samples.

## 2.4. Ångström absorption coefficient

The Ångström exponent was determined from the measurement of the absorbance of an aerosol sample in isopropanol using a UV-Vis spectrometer. The value of Angstrom exponent was then converted for use with the imaginary refractive index as described in the appendix. Samples including humic acid and those obtained from atmospheric sampling may have a measurable wavelength dependent absorption that can be defined by the Ångström exponent. In the context of Mie scattering, absorption is observed as a decrease in spectral intensity of the Mie spectrum. Inclusion of a wavelength

dependent imaginary refractive index term in the Bohren and Huffman (1983) formalism can simulate the attenuation of intensity observed in the Mie spectra owing to absorption.

The absorption Ångström exponent was determined by fitting an Ångström equation (see e.g. Moosmüller et al., 2011) to the absorbance spectra of the atmospheric aerosol extract in isopropanol or humic acid in water obtained by using UV-Vis spectroscopy:

$$\frac{Abs}{Abs_0} = \left(\frac{\lambda}{\lambda_0}\right)^{-\alpha}, \qquad (2)$$

where $Abs$ is the absorbance measured by a UV-Vis spectrometer (Perkin Elmer Lambda 950), $\lambda$ is the wavelength and $\alpha$ is the absorption Ångström exponent. It should be noted that $Abs_0$ is the value of absorbance at the reference wavelength $\lambda_0$=460 nm. The absorbance spectra of the bulk atmospheric aerosol extracts dissolved in isopropanol or the humic acid dissolved in water were recorded with a spectrometer covering the wavelength range 460 to 640 nm. Wood smoke aerosol extract and aqueous humic acid both demonstrated measurable absorption at smaller wavelengths, however the spectra from the other samples were below the instrument detection limits. The quoted photometric noise for the UV-Vis spectrometer was ~0.0002 A. However, the urban and remote aerosol extracts were diluted in isopropanol to fill the UV-Vis spectrometer cuvette and thus a value two orders of magnitude larger than ~0.0002 A may provide an upper bound for the absorbance of the samples reported below the detection limit.

In a UV-Vis spectrometer the absorption coefficient, $\beta$, can be related to the Absorbance, $Abs$, by,

$$Abs = -\beta l \qquad (3)$$

where $l$ is the pathlength (1 cm for the work described here) and absorbance, $Abs$ has been corrected from base 10 to base $e$ (see e.g. Petty, 2006). The absorption coefficient can be related to the imaginary refractive index, $k(\lambda)$, by

$$k(\lambda) = \frac{\lambda\beta}{4\pi} \qquad (4)$$

as described by Petty (2006). Substitution of Eq. 4 into Eq. 3 and subsequently Eq. 2 demonstrates that the Ångström relationship for absorbance (Eq. 2) is modified to

$$\frac{k}{k_0} = \left(\frac{\lambda}{\lambda_0}\right)^{-(\alpha-1)} \qquad (5)$$

for describing the imaginary refractive index (see appendix), where, $k$, represents the imaginary refractive index, $\lambda$, is the wavelength and, $\alpha$, is the absorption Ångström exponent. In essence the value of Ångström exponent, $\alpha$, measured by the UV-Vis spectrometer is larger than the corresponding value for the imaginary refractive index. Note for the work described here $\lambda_0$=460 nm. The values of $k_0$ and $\alpha$ were measured for dilute solutions of the wood smoke extract in isopropanol and humic acid in water. In the optical trap the droplet of wood smoke extract in isopropanol had lost all of the isopropanol

solvent to evaporation, as expected, leaving pure wood smoke extract. The aqueous humic acid solution lost some water to evaporation but remained an aqueous, but more concentrated, solution. As will be described below the mass density of the woodsmoke extract was measured independently. Thus, for the wood smoke extract droplet the values of $k_0$ and $Abs_0$ were corrected for the mass density of wood smoke extract in the optical trap and the attenuation of the resulting Mie spectrum will be shown to be consistent. For the aqueous humic acid solution, the value of $k_0$ was determined by fitting the attenuation of the Mie spectrum by inspection, i.e. by changing the value of $k_0$ until the intensity attenuation of the simulated and experimental Mie spectra matched, thereby calculating the value of the mass density in the trapped humic acid droplet. For the wood smoke aerosol extract, the mass density of the extract in isopropanol was determined from a measurement on an Anton Parr densitometer, whilst the mass density of the pure material was calculated gravimetrically by evaporating isopropanol from a pre-weighed sample of the extract. For the measurement of mass density obtained by the densitometer, a plot of the inverse of the density versus its corresponding weight fraction allowed the mass density of the sample to be determined. The Ångström coefficient determined for the absorbance in isopropanol or water was adjusted for use with the imaginary refractive index Ångström relationship (see Appendix). Simulated Mie spectra of wood smoke aerosol and humic acid aerosol were then calculated with and without application of an Ångström exponent absorption to demonstrate that the attenuation in Mie resonance intensity was consistent with absorption.

## 3. Results

Aqueous humic acid aerosol and extracts of atmospheric aerosols were optically trapped and the real component of the refractive index determined through comparison of experimentally obtained Mie spectra to simulated Mie spectra by varying the radius and the Cauchy coefficients $A$ and $B$. The correct simulation of the Mie spectra requires the variation of absorption with wavelength to be described in terms of the imaginary refractive index, Eq. 5. The Mie spectra for urban and remote atmospheric aerosol extracts are shown in Fig. 1, and the Mie spectra for the wood smoke aerosol extract and aqueous humic acid aerosol are shown in Fig. 2. Typically, small aerosols (with a radius of approximately 0.600 μm) were trapped when studying the atmospheric aerosol extracts and consequently only a few resonances were observed in the Mie spectra depicted in Figs. 1 and 2. Table 1 displays the Cauchy coefficients, refractive index and radius of the aerosols studied. In general, the real component of the refractive index increases from remote to urban to wood smoke.

The mass density of the two pure wood smoke aerosol extracts was determined to be 1.47 g cm$^{-3}$ for extract A and 1.64 g cm$^{-3}$ for extract B. The values are similar to the values Hoffer et al., (2005) reported for humic like substances (HULIS) sourced from a biomass burning plume: 1.502 to 1.569 g cm$^{-3}$ and Dinar et al., (2008) reported 1.42-1.51 g cm$^{-3}$. The mass density of the aqueous humic acid solution in the nebuliser was 0.0005 g cm$^{-3}$, (*i.e.* the mass of humic acid per unit volume of solution) the mass density of the pure humic acid was reported from the supplier Sigma-Aldrich to be 1.52 g cm$^{-3}$. The absorption spectrum and imaginary component of the refractive index over the wavelength range of 460 to 640 nm for the wood smoke and humic acid samples are shown in Fig. 3. Table 2 contains the values of the $k_0$, $Abs_0$, and α determined

in the study presented here. The dependence of the Mie spectral intensity with and without contribution from the absorption are shown for the two absorbing samples in Fig. 2. Absorption attenuates the intensity of the Mie spectra and is most notable at shorter wavelengths where the mass absorption coefficient is largest.

Figure 2 demonstrates that it may be possible to determine the absorption spectra from Mie spectra recorded from optically trapped aerosol. The solvent (isopropanol) was lost from the trapped aerosol extracts by experimental design to ensure the pure extract was studied. In contrast the aqueous humic acid droplet retained some of its water in equilibrium with the local humidity of the optical trapping cell. The trapped droplet was observed to lose water and consequently shrink in size. The simulated Mie spectra in Fig 2. were calculated with and without absorption described by an Ångström exponent to demonstrate that the attenuation of Mie resonances (especially at shorter wavelengths) was consistent with measured Mie spectra. The mass density of humic acid in an aqueous droplet is proportional to the imaginary refractive index, allowing the mass density of the optically trapped humic acid droplet to be calculated. The concentration of the trapped humic acid was determined to be 0.016 g cm$^{-3}$. The concentration of the aqueous humic acid droplet had increased by a factor of ~32 upon trapping, thus demonstrating that water had evaporated from the droplet during the trapping and aerosol equilibration process.

## 4. Discussion

Studies reporting the refractive index of atmospheric aerosol extracts are predominantly conducted at individual wavelengths (see e.g. Guyon et al., 2003; Hoffer et al., 2005; Kim and Paulson, 2013; Lambe et al., 2013; Lang-Yona et al., 2010; Nakayama et al., 2013; Redemann et al., 2000; Stelson et al., 1990). Yamasoe et al. (1998) determined the real component of the refractive index of smoke aerosol extracts to be 1.53, 1.55, 1.59, and 1.58, for wavelengths of 438, 670, 870, and 1020 nm respectively, whilst Shingler et al. (2016) conducted in-situ aerosol particle measurements of wildfire, biogenic, marine and urban air masses and discovered a refractive index of 1.52 to 1.54 at a wavelength of 532 nm. Contrastingly, the refractive index in the study presented here was not calculated at a single wavelength but over a large continuous wavelength range (460 to 700 nm).

### 4.1. Refractive index of atmospheric aerosol extracts

The variation in refractive index between the extracts investigated in the study indicates a distinctive difference between each sample from each location source. Figure 4 graphically compares the refractive index dispersion with wavelength obtained for the atmospheric aerosol extracts analysed in the study presented to selected values from literature. Large differences between the values of refractive index for remote, urban and wood smoke aerosol extracts can be easily observed in Fig. 4. Wood smoke aerosol extracts have the largest values of refractive index, followed by urban and then remote aerosol extracts. Antarctica is considered a clean environment owing to the physical remoteness of the continent and air that reaches Antarctica is considered relatively cleansed of anthropogenic particles (see e.g. Wolff et al., 1990). However, some

particles do reach Antarctica, examples of such aerosol sources include sea spray, the transport of industrial emissions (see e.g. McConnell et al., 2014) and particulate material from biomass burning and tropical forest fires (see e.g. Tomasi et al., 2007). Aerosols from such sources have travelled far and have likely undergone chemical ageing, and are thus likely to be very different in chemical composition than their initial composition. The review authored by Moise et al., (2015)

demonstrates the importance of chemical reactions in the alteration of the optical properties of atmospheric aerosols during atmospheric transport.

The urban aerosol extract samples have a wide distribution of the values of refractive index with values ranging from 1.478 to 1.522 at 589 nm. From the small sample analysed in the study presented, the autumn and winter samples are at the larger end of the refractive index range, with spring and summer at the mid to low end.  The wavelength dependent

refractive index values determined for atmospheric aerosol extracts lie in good agreement with previous monochromatic literature results. Studies focusing on anthropogenic aerosols determined values of refractive index varying from 1.498 to 1.653 nm at 532 nm; the refractive index range found in literature encompasses the urban and wood smoke aerosol extract wavelength dependent refractive indices determined in the study (see e.g. Adler et al., 2011; Yamasoe et al., 1998; Hoffer et al., 2005; Shingler et al., 2016). Previous studies focused on the refractive index of aerosols sourced from Antarctica

demonstrate similar values to those determined in the work presented: Virkkula et al. (2006) measured refractive indexes that generally lay between 1.4 and 1.5, but reached as low as 1.3 (an average of measurements from the wavelengths 450, 550 and 700 nm) for Antarctic aerosol extracts.

Contrastingly, Guyon et al. (2003) studied the refractive index of biomass burning aerosols collected from the Amazon tropical forest and determined a refractive index of 1.41 at 545 nm (0.18 lower than the wavelength dependent

refractive index determined in the study presented here at the same wavelength). A likely reason for the variation is that Guyon et al. (2003) collected samples from a station suspended 54 m above ground level (and 22 m above the forest canopy), whereas in the study presented here samples were drawn directly from the smoke plume from a burning fire and consequently samples did not have the opportunity to chemically age or to mix with other material present in the atmosphere. Refractive index values from literature for secondary organic aerosols demonstrate a wide range of refractive index values

(see e.g. Kim et al., 2010, 2013; Lambe et al. 2013; Lang-Yona et al. 2010; Spindler et al. 2007; Yu et al. 2008). The work here indicates that a single refractive index value cannot be used to describe secondary organic aerosol, perhaps owing to the wide range of organics present in the aerosol (see e.g. Yu et al, 2008).

## 4.2. Refractive index of aqueous humic acid aerosol

Aqueous humic acid was studied to demonstrate the ability of using optical trapping and Mie spectroscopy to study a known

absorbing aerosol. Humic acid was dissolved in water to form a solution prior to nebulisation. The measured refractive index of the aqueous aerosol droplet will have a different refractive index than a pure sample of humic acid owing to the water content of the aerosol. The refractive index of humic-like substances has been reported to be quite large, for example Hoffer et al. (2005) reported a refractive index of 1.653 for HULIS samples. However, in the study presented, the aqueous humic

acid droplets had a refractive index lower than any of the atmospheric aerosol extracts because the atmospheric aerosol extracts were trapped as concentrated aerosol extracts (the solvent, isopropanol, used to allow the aerosol extracts to become airborne evaporated during nebulisation and trapping).

### 4.3. Calculation of the absorption Ångström exponent

Unlike the urban and remote atmospheric aerosol extracts, the wood smoke aerosol extract and aqueous humic acid aerosol had strong absorption properties as shown by UV-Vis experiments on samples dissolved in isopropanol and water respectively. The imaginary component of the refractive index and absorption Ångström exponent could then be determined for the two samples.

Previous studies in the field have determined values for the absorption Ångström exponents to range from 2 to 16 for carbonaceous aerosol over the visible wavelength range (see e.g. Hoffer et al., 2005; Lewis et al., 2008; Chakrabarty et al., 2010; Flowers, 2010; Moosmüller et al., 2011; Utry et al., 2013; Zhang et al.,2013; He et al., 2015; Garg et al., 2016; Pokhrel et al., 2016; Rathod et al., 2016; Shen et al., 2017). More specifically, values in the range of 3.5 to 8.3 have been calculated for studies focusing on the absorption Ångström exponent for smoke aerosols. Lewis et al. (2008) studied the combustion of a variety of fuels with a dual wavelength photo-acoustic instrument to determine an absorption Ångström exponent of 3.5. Contrastingly, Hoffer et al. (2005) and Park and Yu (2016) obtained much larger values of the absorption Ångström exponents of 6 to 7 and 7.4 to 8.3 respectively for biomass burning aerosols.

The calculated absorption Ångström exponent for the wood smoke aerosol extract correlates with the absorption Ångström exponent measured in previous studies for biomass burning aerosols. Interestingly, it has been suggested that fire type plays a role in the amount of black carbon produced. A flaming fire has been shown to produce more particles (see e.g. Reid et al., 2005), and in particular produce more black carbon than smouldering fires (see e.g. Hoffer et al.,2005; Yamasoe et al., 2000). The fire from which the wood smoke aerosol extracts were collected was flaming and hence a high absorption Ångström exponent is expected. It ought to be noted that the wood smoke extract was included in this work as an exploratory sample with strong absorption behaviour and future work will explore smoke aerosol where the fuel and fire temperature are carefully controlled.

Considering the absorption Ångström exponent for the aqueous humic acid aerosol, it can be observed that the exponent correlates with studies that determined the absorption Ångström exponent for biomass fuels. For example, Schnaiter et al. (2006) deduced an absorption Ångström exponent between 2.2 and 3.5 for aerosols produced from the combustion of propane and Schnaiter et al. (2003) determined an absorption Ångström exponent of 1 for emissions produced from the combustion of diesel.

### 4.4. Uncertainty in Mie spectra fitting

The collection technique applied to extract remote and urban aerosols from the atmosphere was limited by airflow and filter size and therefore sample was very limited. Organic material extracted from filters used in high-volume aerosol samplers

were not used in the study presented here as the filter blanks demonstrated contamination. The contamination was attributed to the quality of the filters used, demonstrating that pre-combusted filters are critical.

Owing to limited sample, only small droplets were optically trapped causing the collected Mie spectra to have little structure. A less structured Mie spectrum reduces the accuracy of the determined wavelength dependent refractive index,

radius and absorption Ångström exponent. The Mie spectra in Fig. 1 of summer urban aerosol extracts are structured with pronounced peak shapes that allow the facile fitting between simulated and measured Mie spectra. Such spectra allow a relatively small range of values of the radius, the Cauchy coefficients A, B and C to provide a good fit between measured and simulated Mie spectra. The rest of the Mie spectra in Fig.1 have significantly fewer Mie resonances and their peak shapes are less pronounced. The uncertainties become larger as the spectra become less structured. Despite the limitations in

Mie spectra simulation, the typical uncertainty in radius and refractive index for a Mie spectra was typically ± 6 nm and ± 0.015 respectively, whilst the uncertainty for the absorption Ångström exponent was 7 % for the wood smoke aerosol extract and 5 % for the aqueous humic acid aerosol.

The liquid droplets are assumed to be perfectly spherical. Mie scattering from droplets experiencing small deformation has been shown by Arnold et al. (1990) and Schweiger et al. (1990) to result in resonances which shift, broaden

and split as the droplet asymmetry increases.

The sensitivity of the simulated Mie spectra to the refractive index (±0.015) and radius (±6 nm) of the droplet are shown in Fig. 5. The simulated spectra, with the stated variations, and the experimental Mie spectra for the spring urban aerosol extract are plotted. Figure 5 also contains a third simulated set of Mie spectra calculated by re-optimising the values of A, B, and C in Cauchy equation to achieve a fit between simulated and experimental Mie spectra for particles with a

20 radius ±12 nm from the optimum fit to the experimental data. Fig. 5 demonstrates that the quoted uncertainties in radius (±6 nm) and refractive index (±0.015) are realistic.

### 4.5. Atmospheric Implications

Atmospheric aerosol can increase the top of the atmosphere albedo by scattering incoming solar radiation and decrease the

25 top of the atmosphere albedo by absorbing solar radiation. Using the refractive index data collected in the study presented, a radiative transfer model was applied to consider the change in top of the atmosphere albedo owing to an aerosol film forming with the same optical properties as the extracts studied in the presented study. The material extracted from the atmospheric samples described in the study may form an organic shell at the air-water interface of an aqueous aerosol (see e.g. Gill et al. (1983). An atmospheric radiative-transfer model (see e.g. Stamnes et al., 1988) was applied to study an atmospheric aerosol

layer consisting of core aqueous aerosol coated in an organic shell with the optical properties of the atmospheric aerosol extract measured within the work presented here. The change in the top of the atmospheric albedo was calculated as the proportions of water and organic material were varied for different size aerosols.

The top of the atmosphere albedo was calculated for an aerosol layer with the composition of an aqueous core aerosol surrounded by a shell of either urban, wood smoke or remote atmospheric aerosol extract with the volume fraction varying from 0 to 1 (i.e. pure water to pure organic). The calculations represent only a small-scale study to simply identify potential effects on the top of the atmosphere albedo owing to the presence of pure core-shell particles in the atmosphere versus no aerosol present. The change in the top of the atmosphere albedo is reported as an aerosol relative effect, following the approach of Mishra et al. (2015).

The atmospheric radiative-transfer model uses the DISORT code (see e.g. Stamnes et al., 1988). The model uses values of the scattering, absorption and the asymmetry parameter of aerosols to calculate the change in solar radiation through the atmosphere. To calculate the scattering and absorption parameters for coated spheres Mie calculations were performed for the core-shell particles using BHCOAT, a code developed by Bohren and Huffman (1983), which was later modified to also include a calculation of the asymmetry parameter. Scattering, absorption and asymmetry parameter for the particle are calculated from the refractive index of the core and shell. For all aerosol particles, the refractive index of the core is a wavelength dependent value for water (IAPWS, 1997) and the refractive index of the surrounding medium a wavelength independent value for air of 1.00-0.0i. The shell of the aerosol has a wavelength dependent refractive index of the urban, remote or wood smoke aerosol extracts, as displayed in Table 1. In addition, the absorption properties of the wood smoke aerosol extract were included in the calculation.

The core-shell Mie calculation was used to obtain scattering and absorption cross-sections and asymmetry parameters for particles with a radius of 100 to 10,000 nm (in 100 nm intervals from 100 to 1000 nm, and 1000 nm intervals from 1000 to 10,000 nm), and with the shell volume being a proportion of 0.01 to 0.99 of the whole particle volume. Calculations were performed over wavelengths covering 350 to 750 nm. The ground albedo was set to 0.1. Aerosol of one size was placed in three consecutive 1 km thick layers at the surface, forming a 3 km thick aerosol layer. The aerosol optical depth for each of these layers was set to 0.126, the global average from Mao et al. (2014), and no aerosol or cloud was placed in any subsequent layers. The solar zenith angle was set at 60°. The albedo of the top of the atmosphere was calculated as the ratio of incoming to outgoing irradiance at 100 km altitude and averaged over wavelengths from 350 to 750 nm for each particle size. Calculations were also performed for no aerosol present in the atmosphere. The aerosol radiative effect was then calculated using Eq. 5.

$$ARE_{TOT} = \alpha^{TOA}_{aerosol} - \alpha^{TOA}_{no\ aerosol}, \tag{6}$$

where $ARE_{TOT}$ stands for total aerosol radiative effect, $\alpha^{TOA}_{aerosol}$ stands for the top of the atmosphere albedo with aerosol present and $\alpha^{TOA}_{no\ aerosol}$ stands for the top of atmosphere albedo without aerosol present.

The results of these calculations are presented in Fig. 6. From Fig. 6, it can be observed that all aerosols have a positive effect on the total aerosol radiative effect, with the most positive effect observed for particle sizes of 600 and 800 nm for the urban and remote atmospheric aerosol extracts and 200 and 400 nm for the wood smoke aerosol extract. Note the

change in top of the atmosphere albedo is most pronounced when the volume of shell increases from 0 to 25 % of the total volume.

**5. Conclusions**

A new technique using optical trapping techniques applied alongside Mie spectroscopy was employed to determine the real and imaginary components of the refractive index of insoluble organic material from atmospheric aerosol extracts over a wide wavelength range. The atmospheric aerosol extract was successfully trapped, demonstrating that the material forms spherical liquid droplets to which Mie theory could be applied. From application of Mie theory, the refractive indices of the atmospheric aerosol extract were determined to vary from 1.470 for aerosol extracted from Antarctica to 1.588 for wood smoke aerosol extracts at a wavelength 589 nm, whilst seasonal refractive index dependence was observed for atmospheric aerosol extracted from an urban environment.

Additionally, owing to the efficient light absorbing nature of the wood smoke aerosol extract and aqueous humic acid extract, the absorption Ångström exponent could be determined with a high level of certainty for the extract; through applying optical trapping and Mie spectroscopy alongside UV-Vis spectroscopy it was possible to determine the real and imaginary component of the refractive index.

The aerosol collected may exist as an aerosol in the atmosphere or it may coagulate with other atmospheric aerosols to form a film. Use of a simple one-dimensional radiative-transfer model to study an atmospheric layer of aerosol with a thin shell of atmospheric aerosol extracts on an aqueous spherical core indicates that the albedo at the top of the atmosphere may change by up to 0.03 relative to a pure aqueous droplet.

**Data Availability**

The data presented in the study can be found at http://doi.org/10.5281/zenodo.834450.

**Appendix A**

The absorption Ångström exponent and imaginary refractive index were calculated by using experimentally determined absorption in a series of equations. Equation (2) from Sect. 2.4. is

$$\frac{Abs}{Abs_0} = \left(\frac{\lambda}{\lambda_0}\right)^{-\alpha}, \qquad (A1)$$

where $Abs$ is the absorbance, $\lambda$ is the wavelength and $\alpha$ is the absorption Ångström exponent. It ought to be noted that $Abs_0$ is the value of absorbance at the reference wavelength $\lambda_0$. By substituting Eq. 4 into Eq. 3 and then Eq. A1 the relationship between the two variables was determined:

$$\left(\dfrac{\frac{4\pi k}{\lambda}}{\frac{4\pi k_0}{\lambda_0}}\right) = \left(\dfrac{\lambda}{\lambda_0}\right)^{-\alpha}, \qquad \text{(A2)}$$

$$\left(\dfrac{k}{k_0}\right)\left(\dfrac{\lambda_0}{\lambda}\right) = \left(\dfrac{\lambda}{\lambda_0}\right)^{-\alpha}, \qquad \text{(A3)}$$

$$\left(\dfrac{k}{k_0}\right) = \left(\dfrac{\lambda}{\lambda_0}\right)^{1}\left(\dfrac{\lambda}{\lambda_0}\right)^{-\alpha}, \qquad \text{(A4)}$$

$$\left(\dfrac{k}{k_0}\right) = \left(\dfrac{\lambda}{\lambda_0}\right)^{-(\alpha-1)}, \qquad \text{(A5)}$$

where $\lambda$ is the wavelength, $\alpha$ is the absorption Ångström exponent and $k$ is the imaginary refractive index. Note that $k_0$ is the imaginary refractive index at $\lambda_0$. Equation A5 demonstrates that the relationship between the Ångström exponent of absorbance and Ångström exponent of imaginary refractive index occurs when the Ångström exponent of the imaginary refractive index is $\alpha$-1. Hence, throughout the paper the Ångström exponent is of the imaginary refractive index is $\alpha$-1.

**Author Contributions**

Rosalie H. Shepherd conducted all experiments, extracted all atmospheric aerosol, analysed and interpreted the data collected and wrote the paper. Martin D. King and Andrew D. Ward conceived the experiment and assisted during the experiment. Additionally, Martin D. King collected the urban atmospheric aerosol extracts. Amelia Marks with the assistance of Martin D. King modelled a film of atmospheric aerosol extract on an aqueous aerosol. Neil Brough collected aerosol samples from Antarctica.

**Competing Interests**

The authors declare that they have no conflict of interest.

**Acknowledgments**

The authors would like to thank the Central Laser Facility for granting access time on the optical trapping equipment at Rutherford Appleton Laboratories, Oxfordshire under the grant number 15130028. Rosalie H. Shepherd would like to thank STFC for funding the grant ST\L504279\1.

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

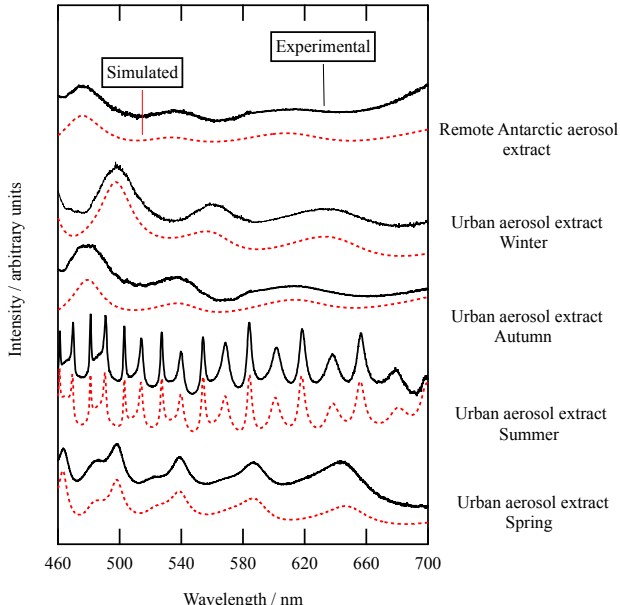

**Figure 1: The typical Mie spectra as a function of wavelength obtained for the urban atmospheric aerosol extracts (collected over the four seasons) and the remote Antarctic atmospheric aerosol extract. From the Mie spectra, the refractive index and radius of the trapped droplet could be determined. Simulated and measured Mie spectra are compared.**

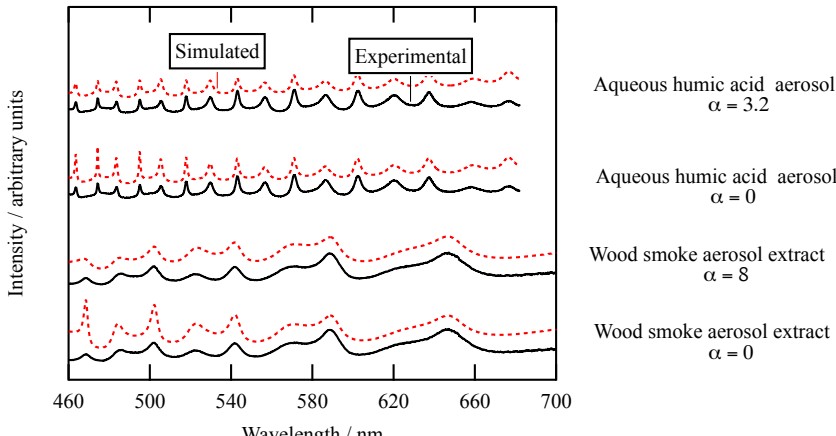

**Figure 2: Typical Mie spectra for the wood smoke aerosol extract B and the humic acid aerosol with simulated Mie spectra with and without absorption. The simulated Mie spectra when the measured absorption Ångström exponent, $\alpha$, was included or when the absorption Ångström exponent was held at zero are shown for wood smoke aerosol extract and humic acid aerosol. Note the absorption measured in Fig. 3 is required to provide a good fit to the intensity of the data, especially at short wavelength.**

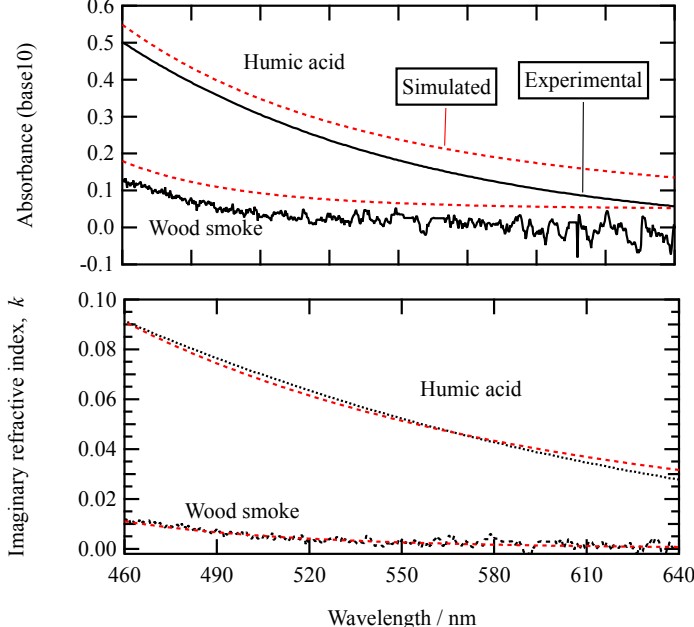

**Figure 3: The absorbance (base 10) of the aqueous humic acid solution and wood smoke aerosol in isopropanol. The mass density of humic acid in water is $7.00\times10^{-5}$ g cm$^{-3}$ and the mass density of the wood smoke extract in isopropanol is $6.60\times10^{-5}$ g cm$^{-3}$. Using mass densities of pure humic acid (1.52 g cm$^{-3}$) and wood smoke extract B (1.64 g cm$^{-3}$) allows calculation of the imaginary refractive index of the pure components as displayed in the second, lower panel. The simulated absorbance curves displayed in the figure are calculated using Eq. (2), using parameters contained in table 2. The simulated absorbance curves in the upper panel have been displaced upwards for clarity.**

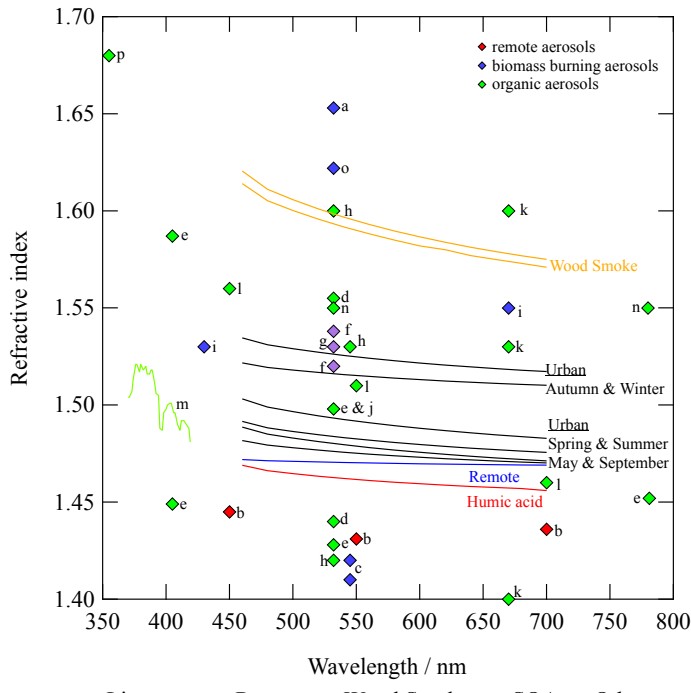

**Figure 4: Refractive index dispersions for urban, remote and wood smoke atmospheric aerosol extracts and humic acid aerosol, compared to refractive index values from literature. A sample of literature studies investigated aerosols from (1) remote locations e.g. b: Virkkula et al. (2006), (2) biomass burning e.g. a: Hoffer et al. (2005), c: Guyon et al. (2003) i: Yamasoe et al. (1998), n: Chakrabarty et al. (2010) and o: Dinar et al. (2008) and (3) organic aerosols e.g. d: Kim and Paulson (2013), e: Nakayama et al. (2013), h: Lambe et al. (2013), j: Spindler et al. (2007), k: Kim et al. (2010), l: Yu et al. (2008), m: Flores et al. (2014), n: Chakrabarty et al. (2010) and p: Trainic et al. (2011).**

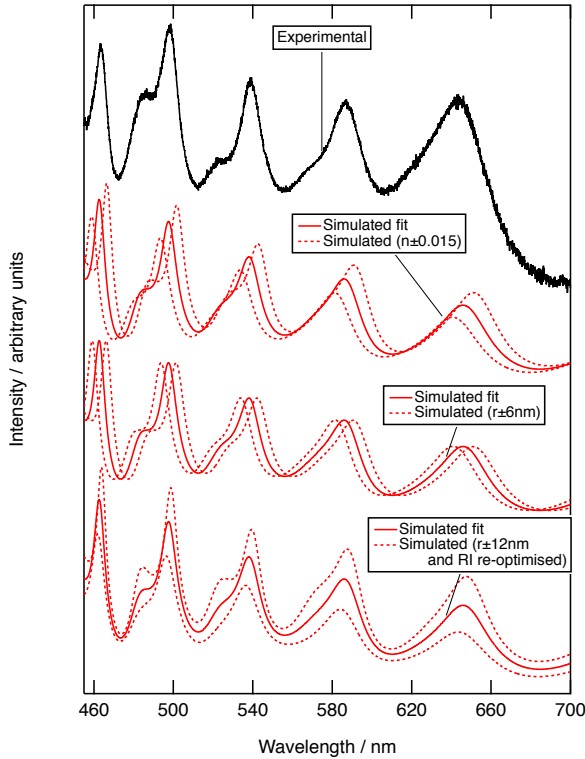

**Figure 5: The experimental Mie spectra for the urban spring aerosol extract with the simulated best fit perturbed in three different scenarios to demonstrate the sensitivity of fitting simulated Mie spectra to experimental Mie spectra. Initially the simulated fit (red solid line) is recalculated with a refractive index increased and decreased by 0.015 (n±0.015), followed by the simulated fit (red solid line) recalculated with a radius increased and decreased by 6 nm (n±6 nm). The final set of simulated Mie spectra consider the simulated fit (red solid line) is recalculated with a droplet radius increased and decreased by 12 nm, but the refractive index re-optimised to get the best fit to the experimental fit (r±12 nm and RI re-optimised). A clear demonstration that the quoted uncertainties in radius (±6 nm) and refractive index (±0.015) are conservative and more than adequate.**

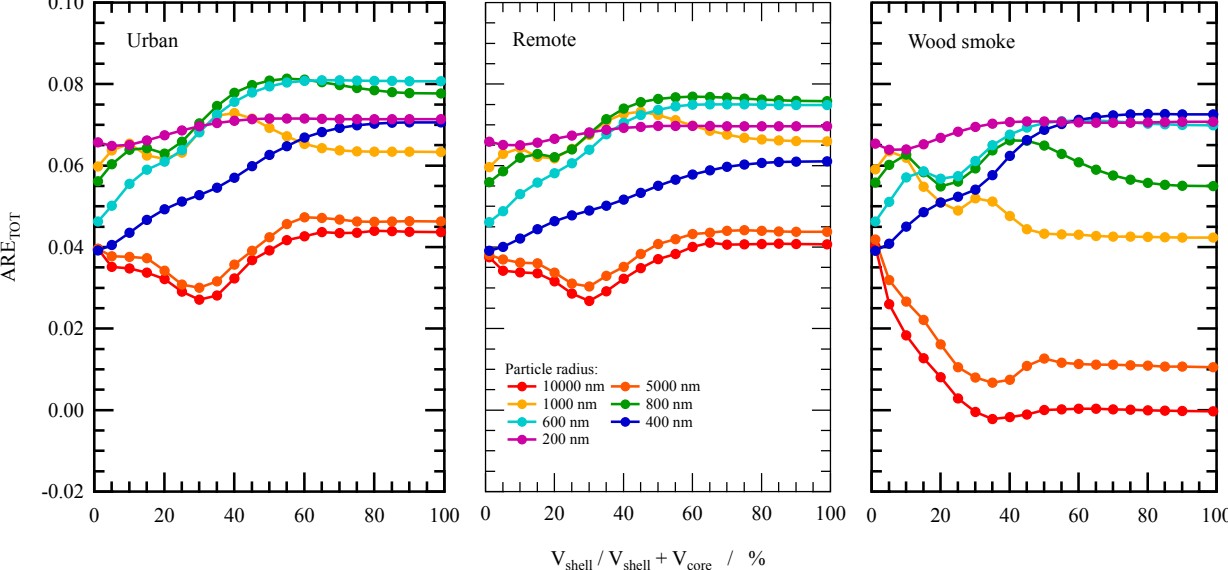

**Figure 6: The aerosol radiative effect for the change in top of the atmosphere albedo upon core-shell morphology formation in atmospheric aerosols. The shell material consists of the urban atmospheric aerosol extract (top), remote atmospheric aerosol extract (middle) or wood smoke atmospheric aerosol extract (bottom), whilst the core for all calculations was water. The radius of the core was varied 7 times. The left-hand side of the graph corresponds to pure aqueous droplets, the right-hand side to pure organic droplets.**

**Table 1: The number of aerosols studied, their determined Cauchy coefficients and the particle sizes. Many more particles were studied but were too small to produce Mie spectra that could be fitted with confidence (i.e. the Mie spectra lacked structure). A very limited amount of sample prevented a larger aerosol becoming optically trapped. Where more than one particle was analysed for a single type of sample, e.g. urban spring, the average and standard deviation of the Cauchy coefficients and real refractive index were reported for the particle studied. The standard deviation does not reflect the uncertainty estimated from the fitting process (which is ±0.015 for the real refractive index) but a spread of values obtained for the few particles studied. The range in particle sizes studied is also reported. Note the wood smoke aerosol extract was plentiful and repeated experiments could be performed.**

| | Sample | Particles Analysed | A | B ($nm^2$) | C ($nm^4$) | Real Refractive Index (589 nm) | Radius Range (μm) |
|---|---|---|---|---|---|---|---|
| Urban | Spring | 3 | 1.478±0.010 | 3750±3250 | 0 | 1.489 | 0.49-0.762 |
| Urban | Summer | 4 | 1.465±0.005 | 4625±1200 | 0 | 1.478 | 0.474-1.252 |
| Urban | Autumn | 1 | 1.505 | 6000 | 0 | 1.522 | 0.492 |
| Urban | Winter | 2 | 1.495±0.007 | 4000 | 0 | 1.507 | 0.515 |
| Remote | Antarctic summer 2015 | 1 | 1.467 | 1000 | 0 | 1.470 | 0.503 |
| Wood Smoke | Extract A | 6 | 1.543 | 15700± 750 | 0 | 1.588± 0.002 | 0.500-0.723 |
| Wood Smoke | Extract B | 6 | 1.541± 0.003 | 14800± 2900 | 0 | 1.584± 0.007 | 0.475-0.593 |
| Test | Humic acid (aqueous) | 1 | 1.449 | 3425 | $1.25\times10^{-8}$ | 1.460 | 1.307 |

**Table 2: The mass densities and absorption properties of the wood smoke and humic acid samples used in the work described here. Note that the mass densities of the wood smoke extract in isopropanol and aqueous humic acid solutions refers to the mass of either woodsmoke extract or humic acid in the volume of isopropanol or water respectively and are not the mass densities of the pure compounds which are also reported in the table.**

| Sample | Mass density / g cm$^{-3}$ | Absorption coefficient, $\beta$, at $\lambda_0$ = 460 nm / cm$^{-1}$ | $k_o$ ($\lambda_0$ = 460 nm) | Abs$_0$ ($\lambda_0$ = 460 nm) | $\alpha$ |
|---|---|---|---|---|---|
| Pure wood smoke extract B | 1.64 | 3033 | 0.0111±0.0010 | - | 9.0±0.1 |
| Wood smoke extract B in isopropanol (Fig. 3 – top panel) | 6.60×10$^{-5}$ | 0.122 | (4.47±0.40)×10$^{-7}$ | 0.122 | 9.0±0.1 |
| Aqueous humic acid (Fig. 3 – top panel) | 7.00×10$^{-5}$ | 1.513 | (4.214±0.38)×10$^{-6}$ | 0.499 | 4.2±0.1 |
| Pure humic acid | 1.52 | 25,000 | 0.092±0.046 | - | 4.2±0.1 |
| Optically trapped aqueous humic acid droplet | 0.016 | 273.2 | (1.00±0.50)×10$^{-3}$ | - | 4.2±0.1 |

