# Peer review of "Determination of the refractive index of insoluble organic extracts from atmospheric aerosol over the visible wavelength range using optical tweezers"

_Atmospheric Chemistry and Physics, 2017_

## Referee Comment (RC1) · Anonymous Referee #2 · 10 Sep 2017

Review of:

"Determination of the refractive index of insoluble organic extracts from atmospheric aerosol over the visible wavelength range using optical tweezers"

In this manuscript, Shepherd et al. present an optical trapping technique combined with white light spectroscopy to measure the real and imaginary parts of the refractive index for samples of insoluble material from ambient aerosol samples. They use the data to estimate the effects of organic film-coated particles in the atmosphere revealing

significant changes in the top of the atmosphere albedo compared to an assumption of equal sized water droplets.

This work is interesting and relevant and should be published once the following minor points are clarified or addressed:

1. In the introduction, references to the work of Reid and coworkers on refractive index measurements should be included.

2. The statement of an "unparalleled level of accuracy" seems too strong given the much greater precision and accuracy achievable using cavity-enhanced Raman scattering in optical tweezers.

3. Was all the material extracted during sonication with chloroform and water? What about components that are insoluble in these solvents?

4. Did the authors observe any artifacts of the sonication process (due to formation of radicals) that indicate further chemistry was occurring and changing the samples? Was this controlled for (i.e. a short duration sonication versus much longer sonication)?

5. Why was the mass concentration of humic acid in the droplets so low? At 30% RH, surely most of the water is lost?

6. The x-axis in figure 5 is confusing – as it reads, if the shell and core volumes are equal, this parameters goes to infinity. Please clarify. The y-axis label does not match the text (should it be ARE_TOT?). Further, in the caption, top, middle and bottom are referenced, but the figure is horizontal.

7. Are the droplets fully spherical in these experiments? What would be the signatures of asphericity and how would this impact the fitting of the data?

---

## Referee Comment (RC2) · Anonymous Referee #3 · 11 Sep 2017

Review of Shepherd et al., 'Determination of the refractive index of insoluble organic extracts from atmospheric aerosol over the visible wavelength range using optical tweezers'.

In this manuscript, the authors present measurements of the complex refractive index of insoluble ambient aerosol matter collected on filters from a range of different environments. Refractive index (RI) measurements were made using a combination of optical trapping, white light spectroscopy and UV-VIS spectroscopy. The retrieved refractive index values were then used to model the effect of core-shell particles coated with the

organic matter on the aerosol radiative effect.

The reported research is original and well presented, and is suitable for publication subject to the corrections/clarifications listed below being addressed:

1) Given that the paper looks at extracting the complex refractive index from optically trapped droplets using Mie theory, the authors should be aware of the following reference and include it in the introduction.

R.E.H. Miles, J.S. Walker, D.R. Burnham and J.P. Reid, 'Retrieval of the Complex Refractive Index of Aerosol Droplets from Optical Tweezers Measurements', Physical Chemistry Chemical Physics 14 (2012) 3037–3047.

2) Page 3, Line 8-9.

The authors state that 'The technique allowed the refractive index to be resolved to within 0.015 over a large wavelength range of 460 to 700 nm'. I take this sentence to mean that the uncertainty in the RI values extracted using this technique is +/- 0.015. However, in Table 1 much reduced uncertainties are given for the samples tested, which correspond instead to the standard deviation in RI values retrieved from multiple droplets. For some of the samples, no uncertainty is given at all. If the precision of the RI measurement is indeed +/- 0.015 as stated in the introduction (and again on page 10, line 22), then this is the error which should be included in Table 1.

3) Page 6, Line 2-4.

The authors state that 'The values of the three empirical constants and the radius of the trapped aerosol were iterated until a good comparison was achieved between the simulated and the experimentally obtained Mie spectrum'

Were the experimental and simulated spectra compared by eye or was a fitting algorithm used? The authors should include a figure which shows how sensitive the simulated spectra are to small changes in the radius and RI, in particular for the more featureless spectra, so the reader can see for themselves the sensitivity of the fit.

4) Page 6, Line 21-22.

The authors should state what the instrument detection limit of the UV-VIS spectrometer was in order to provide an upper bound on the absorbance of the atmospheric samples.

5) Figure 7

In keeping with the description in the text (Equation 4) the y-axis label should be ARE(TOT). Based on the description of the x-variable as the volume fraction of the shell (page 11, lines 2-3) the x-axis label also needs changing to Vshell/(Vshell+Vcore).

6) There are several places in the text which seem to contradict each other as to how the Angstrom exponent was determined ie. from optical trapping or from UV-VIS spectroscopy, or from both. Please could the authors clarify the following explicitly: Were values of the Angstrom exponent extracted from the Mie spectra and from the UV-VIS spectra independently and these values then compared to show they were equivalent? The authors should include somewhere the values of the Angstrom exponent determined from each method, along with associated uncertainties.
* * *

---

## Referee Comment (RC3) · Anonymous Referee #1 · 11 Sep 2017

Interesting experimental work dealing with the characterization of aerosol particles where the imaginary part of the refractive index cannot be neglected.

Questions/comments that should be addressed:

1. To echo what the other two reviewers stated: It seems like some important references regarding the determination of the refractive index of single aerosol particles using Mie theory are missing from the introduction.

2. Were the Mie spectra calculated by integrating over the acceptance angle of the

objective? Please add a sentence or two that qualitatively describes the calculation. On page 3 it is stated "simulating the spectrum with Mie calculations." That is too vague.

3. Can you provide more details on how the size and Cauchy parameters were found. A simple grid search? If so, over what space? Was the imaginary part of the refractive index included in the search space?

4. In Section 4.4, the uncertainty associated with fitting less structured Mie spectra is mentioned. What is the origin of this increased uncertainty when fitting Mie spectra of absorbing particles or submicron particles studied at optical wavelengths? Presumably, Mie theory allows you to calculate the observed spectrum very accurately. Yet, when sharp peaks are absent, the uncertainty in the retrieved parameters increases substantially. A thorough answer to this question is not necessary but a slightly more detailed comment in the text would be helpful.

---

## Author Comment (AC1) · 28 Nov 2017

Reply in the PDF file:-

Please also note the supplement to this comment:
https://www.atmos-chem-phys-discuss.net/acp-2017-693/acp-2017-693-AC1-supplement.pdf
* * *
[Figure]

2017.

*Reply to Interactive comment on* **"Determination of the refractive index of insoluble organic extracts from atmospheric aerosol over the visible wavelength range using optical tweezers"**
*by*
**Rosalie H. Shepherd et al.**
**Anonymous Referee #2**

In this manuscript, Shepherd et al. present an optical trapping technique combined with white light spectroscopy to measure the real and imaginary parts of the refractive index for samples of insoluble material from ambient aerosol samples. They use the data to estimate the effects of organic film-coated particles in the atmosphere revealing significant changes in the top of the atmosphere albedo compared to an assumption of equal sized water droplets.
This work is interesting and relevant and should be published once the following minor points are clarified or addressed:

Thank you.

1. In the introduction, references to the work of Reid and coworkers on refractive index measurements should be included.

The following references have been added:-

R.E.H. Miles, J.S. Walker, D.R. Burnham and J.P. Reid, 'Retrieval of the Complex Refractive Index of Aerosol Droplets from Optical Tweezers Measurements', Physical Chemistry Chemical Physics 14 (2012) 3037–3047.

H.-B. Lin, J.D. Eversole, A.J. Campillo, "Identification of morphology dependent resonances in stimulated Raman scattering from microdroplets", Optics Communications,77(5,6) (1990) 407-410

With the following text added to the paper:

"The use of morphological dependent resonances in Raman Spectra to determine refractive index at a fixed wavelength has been reported by Lin et al. (1990) and references therein and Miles et al. (2012)."

2. The statement of an "unparalleled level of accuracy" seems too strong given the much greater precision and accuracy achievable using cavity-enhanced Raman scattering in optical tweezers.

The statement has been removed and the text now reads:

"Application of the optical trapping technique was successfully employed to determine the refractive index of aerosol over a wide wavelength range".

**Fig. 1.**

**Supplement:**

In this manuscript, Shepherd et al. present an optical trapping technique combined with white light spectroscopy to measure the real and imaginary parts of the refractive index for samples of insoluble material from ambient aerosol samples. They use the data to estimate the effects of organic film-coated particles in the atmosphere revealing significant changes in the top of the atmosphere albedo compared to an assumption of equal sized water droplets.
This work is interesting and relevant and should be published once the following minor points are clarified or addressed:

Thank you.

1. In the introduction, references to the work of Reid and coworkers on refractive index measurements should be included.

The following references have been added:-

R.E.H. Miles, J.S. Walker, D.R. Burnham and J.P. Reid, 'Retrieval of the Complex Refractive Index of Aerosol Droplets from Optical Tweezers Measurements', Physical Chemistry Chemical Physics 14 (2012) 3037–3047.

H.-B. Lin, J.D. Eversole, A.J. Campillo, "Identification of morphology dependent resonances in stimulated Raman scattering from microdroplets", Optics Communications,77(5,6) (1990) 407-410

With the following text added to the paper:

"The use of morphological dependent resonances in Raman Spectra to determine refractive index at a fixed wavelength has been reported by Lin et al. (1990) and references therein and Miles et al. (2012)."

2. The statement of an "unparalleled level of accuracy" seems too strong given the much greater precision and accuracy achievable using cavity-enhanced Raman scattering in optical tweezers.

The statement has been removed and the text now reads:

"Application of the optical trapping technique was successfully employed to determine the refractive index of aerosol over a wide wavelength range".

Note the Reviewer's comment on the greater precision and accuracy using the cavity enhanced Raman scattering in optical tweezers is erroneous as both techniques use the same fundamental principles of light scattering from a sphere. The Miles et al. (2012) study is undertaken with ideal samples where greater precision is possible owing to the ability to generate a spectrum with a lot of structure and allowing a precise fit. The samples presented in this work are atmospheric, very limited in amount and supplies were exhausted within a few minutes of nebulization. In our previous studies (Jones et al., 2013), we have achieved similar precisions to the cavity enhanced Raman scattering values.

The accuracy comment is surprising as the accuracy is ultimately based on the resolution of the spectrographs used which are similar here and in the work of Miles et al. (2012).

3. Was all the material extracted during sonication with chloroform and water? What about components that are insoluble in these solvents?

Two extra references (should) have been included:-

1. Folch J., Lees M., Sloane Stanley G. H. 1957. A simple method for the isolation and purification of total lipides from animal tissues. *J. Biol. Chem.* 226: 497–509.
2. Bligh, E. G. & Dyer, W. J. 1959. A rapid method of total lipid extraction and purification. *Can. J. Biochem. Physiol.* 37: 911–917.

And the text now reads:-

"The organic material was extracted from atmospheric aerosol based on techniques adapted from Folch et al(1957) and Bligh and Dyer (1959) and the refractive…"

Only material that was soluble in chloroform was analyzed. Extraction by chloroform is an accepted method to remove insoluble surface active compounds from complex media. The cited reference for this process, Bligh and Dyer has been cited on ~44,000 occasions.

Material that did not dissolve in either chloroform or water remained on the filter material or as solid detritus that was filtered out of the chloroform. The following text has been added:-

"The filter debris with un-extracted material was discarded."

4. Did the authors observe any artifacts of the sonication process (due to formation of radicals) that indicate further chemistry was occurring and changing the samples? Was this controlled for (i.e. a short duration sonication versus much longer sonication)?

There was no change in chemical properties noted owing to sonication. However checks were performed and the following text has been added:-

"The sonication in the extraction process was not found to change the Langmuir isotherm of atmospheric material at the air-water interface."

5. Why was the mass concentration of humic acid in the droplets so low? At 30% RH, surely most of the water is lost?

The mass concentration of humic acid in the nebulizing solution was kept "low" to prevent potential aerosol heating. As noted in Miles et al (2012) the local RH of a droplet and that measured by the RH probe are frequently different. However the reviewer' query has highlighted an error which we have corrected with the following text:-

"The concentration of the trapped humic acid was determined to be 0.016 $g$ cm$^{-3}$. The concentration of the aqueous humic acid droplet had increased by a factor of ~32 upon trapping, thus demonstrating that water had evaporated from the droplet during the trapping and aerosol equilibration process."

6. The x-axis in figure 5 is confusing – as it reads, if the shell and core volumes are equal, this parameters goes to infinity. Please clarify. The y-axis label does not match the text (should it be ARE_TOT?). Further, in the caption, top, middle and bottom are referenced, but the figure is horizontal.

The x-axis has been fixed.

7. Are the droplets fully spherical in these experiments? What would be the signatures of asphericity and how would this impact the fitting of the data?

The following text has been added

"The liquid droplets are assumed to be perfectly spherical. Mie scattering from droplets experiencing small deformation has been shown[3,4] to result in resonances which shift, broaden and split as the droplet asymmetry increases."

3. S. Arnold, D. E. Spock and L. M. Folan, Opt. Lett., 1990,15, 1111.
4 G. Schweiger, Opt. Lett., 1990, 15, 156.

The formation of an aspherical liquid droplet trapped in a gaseous environment would be somewhat surprising as the interfacial tension responsible for the spherical shape are in the 10's of mN m$^{-1}$ range and far exceed those exerted by the laser trap.

Also, orientation of a spherical particle in an optical trap is independent of the Mie spectrum recorded and thus a rotating spherical particle will not affect the recorded Mie spectrum. However, the orientation of an aspherical particle will give different Mie Spectra depending on orientation. Thus, an aspherical particle will give a changing Mie spectrum. During our previous work with Mie scattering of aerosol liquid droplets we have not encountered such behavior. We have experienced gross asymmetry during collisions of droplets and solid beads (unpublished) which results in complete loss of resonance behavior. Other tell-tale signs to any asphericity of the trapped particles would be an inability to fit the trapped particle to reasonable values of the Cauchy Coefficient as discussed in the new reference, Miles et al. (2012).

---

## Author Comment (AC2) · 28 Nov 2017

Review of Shepherd et al., 'Determination of the refractive index of insoluble organic extracts from atmospheric aerosol over the visible wavelength range using optical tweezers'.
In this manuscript, the authors present measurements of the complex refractive index of insoluble ambient aerosol matter collected on filters from a range of different environments. Refractive index (RI) measurements were made using a combination of optical trapping, white light spectroscopy and UV-VIS spectroscopy. The retrieved refractive index values were then used to model the effect of core-shell particles coated with the organic matter on the aerosol radiative effect.
The reported research is original and well presented, and is suitable for publication

Thank you,

subject to the corrections/clarifications listed below being addressed:
1) Given that the paper looks at extracting the complex refractive index from optically trapped droplets using Mie theory, the authors should be aware of the following refer- ence and include it in the introduction.
R.E.H. Miles, J.S. Walker, D.R. Burnham and J.P. Reid, 'Retrieval of the Complex Refractive Index of Aerosol Droplets from Optical Tweezers Measurements', Physical Chemistry Chemical Physics 14 (2012) 3037–3047.

The following references have been added:-

R.E.H. Miles, J.S. Walker, D.R. Burnham and J.P. Reid, 'Retrieval of the Complex Refractive Index of Aerosol Droplets from Optical Tweezers Measurements', Physical Chemistry Chemical Physics 14 (2012) 3037–3047.

H.-B. Lin, J.D. Eversole, A.J. Campillo, "Identification of morphology dependent resonances in stimulated Raman scattering from microdroplets", Optics Communications, 77(5,6), (1990) 407-410

With the following text added to the paper:

"The use of morphological dependent resonances in Raman Spectra to determine refractive index at a fixed wavelength has been reported by Lin et al. (1990) and references therein and Miles et al. (2012)."

2) Page 3, Line 8-9.

The authors state that 'The technique allowed the refractive index to be resolved to within 0.015 over a large wavelength range of 460 to 700 nm'. I take this sentence to mean that the uncertainty in the RI values extracted using this technique is +/- 0.015. However, in Table 1 much reduced uncertainties are given for the samples tested, which correspond instead to the standard deviation in RI values retrieved from multiple droplets. For some of the samples, no uncertainty is given at all. If the precision of the RI measurement is indeed +/- 0.015 as stated in the introduction (and again on page 10, line 22), then this is the error which should be included in Table 1.

The text in the caption for table 1 has been edited to be clearer:-

"Where more than one particle was analyzed for a single type of sample, e.g. urban spring, the average and standard deviation of the Cauchy coefficients and real refractive index were reported for the particle studied. The standard deviation does not reflect the uncertainty estimated from the fitting process (which is ±0.015 for the real refractive index) but a spread of values obtained for the few particles studied. The range in particle sizes studied is also reported."

The distinction is important as the purpose of table 1 is to demonstrate the number of experiments that were performed (as demonstrated by the first line of the caption "Table described the number of aerosols studied, their determined...") and the range of values measured. The

3) Page 6, Line 2-4.
The authors state that 'The values of the three empirical constants and the radius of the trapped aerosol were iterated until a good comparison was achieved between the simulated and the experimentally obtained Mie spectrum'
Were the experimental and simulated spectra compared by eye or was a fitting algorithm used?

The following text has been added:-

"Typically, the radius of droplet was fixed and the values of A, B, and C varied until a good fit between measured and simulated Mie spectra was achieved by simple comparison (inspection) of peak, trough and inflection point positions. The value of the radius was then iterated through a series of radii with optimization of the values of A, B. and C as a function of wavelength repeated at each radius. Thus, a qualitative grid search was performed over parameter space. Parameter space was A varying from 1.3 to 1.7, B from 0 to 20,000 $nm^{-2}$ and C from 0 to $1\times10^9$ $nm^4$. The value of the radius was between 0 to 3 microns typically. The imaginary component of the refractive index was varied only after the grid search for the woodsmoke and humic acid samples shown in fig. 2."

The authors should include a figure which shows how sensitive the simulated spectra are to small changes in the radius and RI, in particular for the more featureless spectra, so the reader can see for themselves the sensitivity of the fit.

The following text and figures have been added to the paper:-

"The sensitivity of the simulated Mie spectra to the refractive index (±0.015) and radius (±6nm) of the droplet are shown in figure 5. The simulated spectra, with the stated variations and the experimental Mie spectra for the Spring Urban aerosol extract are plotted. Figure 5 also contains a third simulated set of Mie spectra calculated by re-optimizing the values of A, B, and C in the Cauchy equation to achieve a fit between simulated and experimental Mie spectra for particles with a radius ±12 nm from the optimum fit to the experimental data. Figure 5 demonstrates that the quoted uncertainties in radius (±6nm) and refractive index (±0.015) are realistic."

[Figure]

(New)"Figure 5: The experimental Mie spectra for the urban spring aerosol extract with the simulated best fit perturbed in three different scenarios to demonstrate the sensitivity of fitting simulated Mie spectra to experimental Mie spectra. Initially the simulated fit (red solid line) is recalculated with a refractive index increased and decreased by 0.015 (n±0.015), followed by the simulated fit (red solid line) recalculated with a radius increased and decreased by 6nm (n±6nm). The final set of simulated Mie spectra consider the simulated fit (red solid line) is recalculated with a droplet radius increased and decreased by 12nm, but the refractive index re-optimized to get the best fit to the experimental fit (r±12nm and RI re-optimized). A clear demonstration that the quoted uncertainties in radius (±6nm) and refractive index (±0.015) are conservative and more than adequate."

4) Page 6, Line 21-22.
The authors should state what the instrument detection limit of the UV-VIS spectrometer was in order to provide an upper bound on the absorbance of the atmospheric samples.

The baseline flatness of Perkin Elmer lambda 950 is quoted as ±0.0008A with a photometric reproducibility of 0.00016A and a RMS photometric noise of 0.0002A. The following text has been added

"The quoted photometric noise for the UV-Vis spectrometer was ~0.0002A. However, the urban and remote aerosol extracts were diluted in isopropanol to fill the UV-Vis spectrometer cuvette and thus a value three orders of magnitude larger than ~0.0002A may provide an upper bound for the absorbance of the samples reported below the detection limit."

5) Figure 7
In keeping with the description in the text (Equation 4) the y-axis label should be ARE(TOT). Based on the description of the x-variable as the volume fraction of the shell (page 11, lines 2-3) the x-axis label also needs changing to Vshell/(Vshell+Vcore).

The x-axis has been fixed.

6) There are several places in the text which seem to contradict each other as to how the Angstrom exponent was determined ie. from optical trapping or from UV-VIS spectroscopy, or from both. Please could the authors clarify the following explicitly: Were values of the Angstrom exponent extracted from the Mie spectra and from the UV-VIS spectra independently and these values then compared to show they were equivalent? The authors should include somewhere the values of the Angstrom exponent determined from each method, along with associated uncertainties.

The Angstrom exponent was determined from the measurement of the absorbance of an aerosol sample in isopropanol using a UV-Vis spectrometer. The value of Angstrom exponent was then adjusted (converted) for use with the imaginary refractive index as described in the appendix. The simulated Mie Spectra in Fig.2 were then calculated with and without an Angstrom component to demonstrate that the attenuation of the Mie Resonances (especially at the shorter wavelengths) was consistent with the measured Mie spectra (in figure2). The following text has been added:-

"The Angstrom coefficient determined for the Absorbance in isopropanol or water was adjusted for use with the imaginary refractive index Angstrom relationship (see Appendix). Simulated Mie spectra of wood smoke aerosol and humic acid aerosol were then calculated with and without an Angstrom exponent

absorption to demonstrate that the attenuation in Mie resonance intensity was consistent with absorption as shown in figure 2."

and

"The simulated Mie spectra in Fig 2. were calculated with and without absorption described by an Angstrom exponent to demonstrate that the attenuation of Mie resonances (especially at shorter wavelengths) was consistent with measured Mie spectra."

and

"In a UV-Vis spectrometer the absorption coefficient, $\beta$, can be related to the Absorbance, Abs, by,

$$Abs = -\beta l \qquad (3)$$

where l is the pathlength (1 cm for the work described here) and absorbance, Abs has been corrected from base 10 to base e (Petty 2006). The absorption coefficient can be related to the imaginary refractive index, $k(\lambda)$, by

$$k(\lambda) = \frac{\lambda \beta}{4\pi} \qquad (4)$$

as described by Petty (2006). Substitution of equation (4) into equation (3) and subsequently equation (2) demonstrates that the Ångström relationship for absorbance (equation 2) is modified to

$$\frac{k}{k_0} = \left(\frac{\lambda}{\lambda_0}\right)^{-(\alpha-1)} \qquad (5)$$

for describing the imaginary refractive index (see appendix). In essence the value of Ångström exponent, $\alpha$, measured by the UV-Vis spectrometer is larger than the corresponding value for the imaginary refractive index. Note for the work described here $\lambda_0$=460nm. The values of $k_0$ and $\alpha$ were measured for dilute solutions of the wood smoke extract in isopropanol and humic acid in water. In the optical trap the trapped droplet of wood smoke extract in isopropanol lost all of the isopropanol solvent to evaporation as expected, leaving pure wood smoke extract. The aqueous humic acid solution lost some water to evaporation, but remained an aqueous and more concentrated, solution. As will be described below the mass density of the woodsmoke extract was measured independently. Thus, for the wood smoke droplet the values of $k_0$ and $Abs_0$ were corrected for the mass density of wood smoke extract in the optical trap and the attenuation of the resulting Mie spectrum will be shown to be consistent. For the aqueous humic acid solution, the value of $k_0$ was determined by fitting the attenuation of the Mie spectrum by inspection, *i.e.* by changing the value of $k_0$ until the intensity

attenuation of the simulated and experimental Mie spectra matched, thereby calculating the value of the mass density in the trapped humic acid droplet."

The values and uncertainties for the Angstrom coefficient are reported in a new table 2.

| Sample | Mass density / g cm$^{-3}$ | Absorption coefficient, $\beta$, at $\lambda_0$ = 460 nm / cm$^{-1}$ | $k_0$ ($\lambda_0$ = 460 nm) | Abs$_0$ ($\lambda_0$ = 460 nm) | $\alpha$ |
|---|---|---|---|---|---|
| Pure Wood smoke extract B | 1.64 | 3033 | 0.0111±0.0010 | - | 9.0±0.1 |
| Wood smoke extract B in isopropanol (Fig. 3 – top pane) | 6.60×10$^{-5}$ | 0.122 | (4.47±0.40)×10$^{-7}$ | 0.122 | 9.0±0.1 |
| Aqueous Humic acid (Fig. 3 – top pane) | 7.00×10$^{-5}$ | 1.513 | (4.214±0.38)×10$^{-6}$ | 0.499 | 4.2±0.1 |
| Pure Humic acid | 1.52 | 25,000 | 0.092±0.046 | - | 4.2±0.1 |
| Optically trapped aqueous Humic acid droplet | 0.016 | 273.2 | (1.00±0.50)×10$^{-3}$ | - | 4.2±0.1 |

---

## Author Comment (AC3) · 28 Nov 2017

Interesting experimental work dealing with the characterization of aerosol particles where the imaginary part of the refractive index cannot be neglected.

Thank you

Questions/comments that should be addressed:
1. To echo what the other two reviewers stated: It seems like some important references regarding the determination of the refractive index of single aerosol particles using Mie theory are missing from the introduction.

The following references have been added:-

R.E.H. Miles, J.S. Walker, D.R. Burnham and J.P. Reid, 'Retrieval of the Complex Refractive Index of Aerosol Droplets from Optical Tweezers Measurements', Physical Chemistry Chemical Physics 14 (2012) 3037–3047.

H.-B. Lin, J.D. Eversole, A.J. Campillo, "Identification of morphology dependent resonances in stimulated Raman scattering from microdroplets", Optics Communications,77(5,6) (1990) 407-410

With the following text added to the paper:

"The use of morphological dependent resonances in Raman Spectra to determine refractive index at a fixed wavelength has been reported by Lin et al. (1990) and references therein and Miles et al. (2012)."

2. Were the Mie spectra calculated by integrating over the acceptance angle of the objective? Please add a sentence or two that qualitatively describes the calculation. On page 3 it is stated "simulating the spectrum with Mie calculations." That is too vague.

Yes, the following text has been changed to read:-

"The optically trapped aerosol droplet was illuminated with white light and the elastically backscattered light was collected over a 25-degree cone angle, by an

objective lens. Further optics (described in Jones et al, 2013) focused the light onto a spectrometer (Acton, SP1500i)."

and

"The measured Mie spectrum was simulated through the application of Bohren and Huffman(1983) formalism of Mie theory integrating over 25 degree cone angle of backscattered light. In conjunction with the Cauchy Equation…"

3. Can you provide more details on how the size and Cauchy parameters were found. A simple grid search? If so, over what space? Was the imaginary part of the refractive index included in the search space?

The following text has been added:-

"Typically, the radius of droplet was fixed and the values of A, B, and C varied until a good fit between measured and simulated Mie spectra was achieved by simple comparison (inspection) of peak, trough and inflection point positions. The value of the radius was then iterated through a series of radii with optimization of the values of A, B. and C as a function of wavelength repeated at each radius. Thus, a qualitative grid search was performed over parameter space. Parameter space was A varying from 1.3 to 1.7, B from 0 to 20,000 $nm^{-2}$ and C from 0 to $1 \times 10^9$ $nm^4$. The value of the radius will be between 0 to 3 microns typically. The imaginary component of the refractive index was varied only after the grid search for the woodsmoke and humic acid samples."

The Imaginary part of the refractive index was described by an Ångström equation. The value of the Ångström exponent was calculated from the UV-Vis Absorbance measurement of the bulk sample and shown to be consistent with the attenuated Mie Spectra in figure 2. The issue and changes to text are further addressed in our answer to point 6, from reviewer 3.

4. In Section 4.4, the uncertainty associated with fitting less structured Mie spectra is mentioned. What is the origin of this increased uncertainty when fitting Mie spectra of absorbing particles or submicron particles studied at optical wavelengths? Presumably, Mie theory allows you to calculate the observed spectrum very accurately. Yet, when sharp peaks are absent, the uncertainty in the retrieved parameters increases substantially. A thorough answer to this question is not necessary but a slightly more detailed comment in the text would be helpful.

The following text has been added:-

"The Mie spectra in figure 1 of Summer Urban aerosol extract are structured with pronounced peak shapes that allow the facile fitting between simulated and

measured Mie Spectra. Such spectra allow a relatively small range of values of radius, Cauchy coefficients A, B and C to provide a good fit between measured and simulated Mie Spectra. The rest of the Mie spectra in Figure 1 have significantly fewer Mie resonances and their peak shapes are less pronounced. The uncertainties become larger as the spectra become less structured."

---

## Author Response (AR2)

Reply to Editor's comments.

*Text from Reviewer/Editor is red coloured.*
Text from Authors is blue coloured.

*Thank you for the revised version. As you will see by the comments of reviewer 2 and 3 and by my own comments below, there are still plenty of technical corrections and clarifications to be done before the manuscript can finally be published in ACP. The current manuscript is not in the form to proceed. A thorough (!) editorial read is inevitable. Please have a look at the manuscript preparation guidelines before adding the technical corrections:* `https://www.atmospheric-chemistry-and-physics.net/for_authors/manuscript_preparation.html`. *Detailed comments:*

- *Table 2: There are a few typos here (e.g. 'pane'). Please double-check the density of the humic acid droplet, which seems quite unreasonable.*
  "Panel" has beens substituted for "pane", although pane is a legitimate english word for a sub-section of a figure or graph.
  The mass densities are correct and reasonable. The following text has been added to the table caption to explain:
  *"Note that the mass densities of the wood smoke extract in isopropanol and aqueous humic acid solutions refers to the mass of either woodsmoke extract or humic acid in the volume of isopropanol or water respectively and are not the mass densities of the pure compounds which are also reported in the table.".*

- *In the abstract, check that the units are correctly set (incl. space between number and unit)*
  The units in abstract have been corrected.

- *Check that all acronyms are properly defined (e.g. DPPC on page 2, line 22 is not defined).*
  DPPC (1,2-dipalmitoyl-sn-glycero-3-phosphocholine )and any other undefined acronyms have now been defined.

- *The citation style is often not properly done. For example, inline citations often end with two parenthesis in a row (see line 27 or 18 on page 2 or line 12, page 4). Within sentences, please put the 'e.g.' inside the parentheses. For example, '...in conjunction with Mie spectroscopy e.g. Bohren and Huffman (1983).' should be '...in conjunction with Mie spectroscopy (see e.g. Bohren and Huffman, 1983).'*
  The citation style has been adopted throughout the paper.

- *Page 2, line 6-8: Please rephrase.*
  The lines have been re-phrased and now reads
  *"The current understanding of the atmospheric aerosol radiative forcing and the cloud albedo effect is currently regarded as low compared to other radiative effects such as greenhouse gases(see e.g. Stocker et al., 2013; Fuzzi et al., 2005). Atmospheric aerosols contain a complex mixture of many different chemical compounds with a wide variety of physio-chemical properties (see e.g. Cappa et al., 2011; Cai et al., 2016; Cochran et al., 2016). "*

- *Throughout the text: Please use the proper unit for 'percent', 'degree', 'microns', etc.*
  Throughout "percent" has been replaced by %, degree by °, and micron by $\mu$m.

- *First sentence of page 6 (before Eq. 1): Please rephrase and check for correct punctuation.*
  The sentence has been broken down into smaller sentences for clarity and now reads *"The measured Mie spectrum was simulated using the computational methods of Bohren and Huffman (1983); integrating over a cone angle of backscattered light of 25°,. The simulated Mie spectrum was calculated as a function of wavelength, with wavelength dependence of the refractive index being described by a Cauchy equation:"*.

- *Page 6, line 11: Please double-check the units of the Cauchy coefficients.*
  Corrected.

- *Page 6, line 22: It should be 'by fitting the Ångström equation'. Or are there different versions of the Ångström equation? If so, please clarify.*
  Corrected as suggested.

- *Throughout the manuscript: There should be a space between number and unit. Please also check the correct abbreviations and correct comma placements!*
  Spaces inserted between numbers and units. Abbreviations checked and comma placements corrected.

- *Please harmonize the usage of 'UV-VIS' and 'UV-Vis'.*
  All occurrences of "UV-VIS" converted to "UV-Vis".

- *Page 8: Equation 4 has already been mentioned and shown on page 7. Again, you need to thoroughly revise your manuscript to exclude further potential sloppiness!*
  Equations renumbered and only appear once in the paper.

- *Page 10, line 5: '0.1812' should only be given with two digits after the comma.*
  Now reported as 0.18.

- *Page 11, second half: Many inconsistencies in capitalization of certain words (figure, Summer, Urban, etc), please revise.*
  Figure now either "Fig." or "Figure" as per instructions. The names of the samples, *e.g.* Summer, Winter *etc.*are now lower case.

- *Page 12, line 2: Add 'the' before 'Cauchy'.*
  Word added.

- *The abbreviation of the variables in Eq. 5 is not very fortunate. If you find a better variable for albedo (which is main parameter here), e.g. omega, then 'TOA' and 'no aerosol'/'aerosol' should be in the superscript and subscript, respectively.*
  As suggested the equation is changed to $ARE_{TOT} = \alpha_{aerosol}^{TOA} - \alpha_{no\ aerosol}^{TOA}$ .

- Page 14, line 14: I can?t find $\sigma$ in the equations above. Please revise.
  Word added.

- *Figure 1: The first lines in the graph (remote Antarctic aerosol extract) is not mentioned in the caption.*
  The word "Antarctic" is added after the word "remote" in the figure caption to be clear.

- *Caption of Fig. 2: The panel labels (a) and (b) are not shown in the actual figure.*
  The "(a)" and the "(b)" have been removed from the figure caption as they are superfluous.

- *Table 1 on page 30: I would recommend to replace 'Aerosols Analyzed' by 'Particles Analyzed' since you only target the particle and not the gas phase.*
  "Aerosols Analyzed" has been replaced by "Particles Analysed".

- Page 30 and 31: The table labeling is wrong. Table 1 exists two times. Please double-check that the labeling is also correct for the figures and tables within the text.
  The second table and all references to it are labelled as table 2.

Reply to Referee 2, report 1.

*Text from Reviewer/Editor is red coloured.*
Text from Authors is blue coloured.

*I thank the authors for acknowledging my comments and making changes accordingly. My comment on the use of Raman scattering in optical tweezers pertained to the angular-independence of whispering gallery modes, as compared to the interference pattern observed in the present study. I acknowledge that in the size range the authors are working that cavity-enhanced resonances will not be possible to detect, and thus my point is moot.*

*The following are minor technical points:*

- *Equation A2 should have lambda_0 in the denominator of the denominator on the left hand side.*
  Corrected. $\lambda_0$ has been added to the denominator.

- *The wording on Line 17, Page 14 needs correcting.*
  The sentence has been rephrased to "The data presented in the study can be found at http://doi.org/10.5281/zenodo.834450.".

- *Equation number is off in the main text due to inclusion of additional equations (eq. 4 features on page 7 and 8, and the page 8 version is the same as eq. 5 on the previous page).*
  Corrected. All equations have been re-numbered..

- *Some inconsistencies in American vs British English (particularly nebulizer vs nebuliser)*
  The remaining "-ize" changed to "-ise".

- *Page 7 line 1: spelling of measurable, page 11 line 4: spelling of respectively.*
  Both spellings corrected.

Reply to Referee3, report 2

*Text from Review/Editor is red coloured.*
Text from Authors is blue coloured.

*Re-review of Shepherd et al.*

- *I recommend the manuscript for publication subject to the following minor corrections.*
  Thank you.

- *Page 6, line 11: The authors should give some indication of the step sized used when varying radius and A, B and C values.*
  The step size used was 0.0001. The following text was added *"(in steps of* $\Delta A = 0.0001$, $\Delta B = 25\,nm^2$*, and* $\Delta C = 5 \times 10^6\,nm^4$*). "* .

- *Page 6, line 13: The units of B are nmˆ2 not nmˆ-2.*
  Corrected. Thank you for spotting this.

- *Section 2.4: Changes have been made here following the comment in my initial review, but I still find the text to be confusing. It would be very helpful if the authors inserted the text they used in their response directly to me in to the manuscript at the start of the section (copied below) as this was very clear.*
    *'The Angstrom exponent was determined from the measurement of the absorbance of an aerosol sample in isopropanol using a UV-Vis spectrometer. The value of Angstrom exponent was then adjusted (converted) for use with the imaginary refractive index as described in the appendix.'*
  The above text has been inserted at the beginning of section 2.5 as requested.

- *Whole manuscript: equations need renumbering as there are now two equation (4)?s and two equation (5)?s.*
  Corrected. All equations have been re-numbered.

- *Page 13 and Figure 6: In the text the y-axis is still labelled ARETOT but on the figure it is ARETOA*
  Corrected. The axis label in Figure 6 has been fixed.

- *Equation A2: It should be lamda0 in the denominator on the left hand side.*
  Corrected. $\lambda_0$ has been added to the denominator.

- *Page 15, line 3: There is no sigma in the equations.*
  Corrected. Definition of $\sigma$ has been removed from the text.

- *Figure 5: Please try and place the label boxes so that they don?t obscure the peaks and troughs in the spectra.*
  Figure 5 has been redrawn. Reducing the text font size and movement of the text boxes has achieved the reviewer's request.

- *Page 32: The new table should be labelled Table 2.*
  Corrected. Second table now labelled as Table 2. .

[revised manuscript text omitted]

Column Break